# Structural plasticity for neuromorphic networks with electropolymerized dendritic PEDOT connections

Kamila Janzakova[1,5], Ismael Balafrej [2,3,5], Ankush Kumar [1,5], Nikhil Garg[1,2,4], Corentin Scholaert[1], Jean Rouat [2,3], Dominique Drouin [2,4], Yannick Coffinier[1], Sébastien Pecqueur [1] & Fabien Alibart [1,2,4] ✉

Neural networks are powerful tools for solving complex problems, but finding the right network topology for a given task remains an open question. Biology uses neurogenesis and structural plasticity to solve this problem. Advanced neural network algorithms are mostly relying on synaptic plasticity and learning. The main limitation in reconciling these two approaches is the lack of a viable hardware solution that could reproduce the bottom-up development of biological neural networks. Here, we show how the dendritic growth of PEDOT:PSS-based fibers through AC electropolymerization can implement structural plasticity during network development. We find that this strategy follows Hebbian principles and is able to define topologies that leverage better computing performances with sparse synaptic connectivity for solving non-trivial tasks. This approach is validated in software simulation, and offers up to 61% better network sparsity on classification and 50% in signal reconstruction tasks.

Replicating brain computing principles remains today an unachievable puzzle, in spite of the recent progresses in our understanding of the various mechanisms at work in the brain machinery. Even if machine learning has reached several milestones in terms of performances, surpassing human level scores on tasks such as image recognition or strategic game playing, our ability to reproduce a true generalized artificial intelligence is still lacking. This is pointing toward the fact that there are still some missing ingredients to add to existing bio-inspired computing solutions in order to reach the next level. Without a complete computational model of the problem in hand, reverse engineering of the brain is an attractive option to identify some of the missing pieces of the puzzle. Toward this quest, neuromorphic engineering and computing have extensively capitalized on bio-inspiration for defining the key components of hardware implementation and algorithms for learning. On the one hand, material implementation of neurons and synapses has been proposed based on conventional complementary metal-oxide-semiconductor (CMOS) technology[1] and emerging nanotechnologies[2,3], with various degrees of resemblance with their biological counterpart. Notably, synaptic plasticity implementation has been revolutionized by the concept of memristor and crossbar integration to enable ultra-high density of synaptic connections. On the other hand, learning algorithms have been deeply influenced by Hebbian learning and backpropagation derivatives to provide efficient bio-inspired synaptic plasticity rules[4–6]. But if neuromorphic engineering is today recognized as a viable computing paradigm that could bring machine learning to its next level, notably in terms of energy efficiency, we are still lacking a clear strategy for defining neural networks topologies, which is realized mostly through costly optimization rules[7]. This issue becomes even more critical when considering the hardware substrate to implement a given algorithm, since nodes and connections of the network have to be mapped on physical devices and components. It turns out that hardware

[1]Univ. Lille, CNRS, Centrale Lille, Univ. Polytechnique Hauts-de-France, UMR 8520-IEMN, 59000 Lille, France. [2]Institut interdisciplinaire d'innovation technologique (3IT), Université de Sherbrooke, Sherbrooke, QC J1X0A5, Canada. [3]NECOTIS Research Lab, faculté de génie électrique et informatique, Université de Sherbrooke, Sherbrooke, QC J1K2R1, Canada. [4]Laboratoire Nanotechnologies & Nanosystèmes (LN2), CNRS, Université de Sherbrooke, Sherbrooke, QC J1X0A5, Canada. [5]These authors contributed equally: Kamila Janzakova, Ismael Balafrej, Ankush Kumar. ✉e-mail: fabien.alibart@univ-lille.fr

substrates have to be over-sized in order to fit any topologies, which are a priori unknown. This aspect impacts directly the power efficiency of neuromorphic hardware and is a strong limitation toward the development of ultra-low power neuromorphic circuits.

In fact, this question is pointing out a fundamental difference between biological neural networks and artificial ones. The hardware implementation of ANNs follows a top-down approach imposed by conventional technology, meaning that topologies need to be a priori known. On the opposite, biology is largely bottom-up. Homogeneous cells differentiate into neurons and project their dendritic connections during network genesis to reach a topology that provides the organism with its specific abilities[8]. This structural plasticity is one of the keys to biological efficiency in terms of (i) energy since material resources are employed only when required and (ii) computational performances since it provides the architectural foundation for the expression of all cognitive tasks occurring during lifetime[9]. In this article, we show how organic materials and devices can reproduce this feature and how bottom-up engineering of ANNs can be used to find topologies with better computing performances. We use bipolar electropolymerization of conducting dendritic PEDOT:PSS fibers in water-based electrolyte to engineer neurons' genesis. We propose Hebbian-like scenarios for dendritic growth and network topologies definition in associative memory, classification and auto-encoding tasks. We show that engineering structural plasticity allows for finding effective topologies for these computing tasks with a massive reduction in connection density.

Finding the right topology of ANNs for a given application remains today an open question. Biology has coped with this issue through millions of years of evolution to encode in the genome of living species the optimal network topologies[10]. Nevertheless, such genetic evolution is not applicable to current deep ANNs and becomes quickly too costly in terms of time and computing resources for large networks[11]. In addition, genotypic approach are not considering the influence of the environment on the network development, which is believed to play a key role in the definition of optimal topologies[12–14]. Such phenotypic plasticity helps the network to find the right topology based on activity induced by the external environment and to leverage resource utilization with higher performances. The lottery ticket hypothesis states that every ANN contains a subnetwork of pruned weights that can achieve results comparable to the initial dense representation, even if trained from random weights, as long as the connectivity matrix is known[15]. Connectivity pruning techniques are not hardware friendly since the starting network before learning needs to be over-sized and connections are removed during or after learning. An attractive option to relax the topologies definition requirement has been to consider random topologies for computing, such as in reservoir computing approaches[16]. However, all reservoirs do not perform equivalently well, and topologies have a strong impact on their performances. Various rules have optimized reservoirs by providing constrained topologies (small world topologies) reflecting the topologies of biological networks[17]. Others have focused on adding local learning rules at the synaptic level in order to adapt the reservoir to its task, which corresponds to translating the topology search space at the synaptic plasticity level[18]. But, there is evidence pointing toward the fact that structural plasticity, or wiring of the network, can bear more profound memory effects[19] and noise resilience[20] than synaptic plasticity does. It is still a missing piece of the puzzle to know how to incorporate structural plasticity into hardware implementations of ANNs. Answering this question cannot be straightforward, and one needs to reconsider profoundly what hardware implementations and technologies should be used to this end.

Interestingly, organic materials and devices have been proposed recently with the ability to realize truly evolvable structures[21]. Bipolar AC electropolymerization of monomers has been investigated in the context of conductive dendritic polymer fibers formation[22,23].

First works have demonstrated that dendritic PEDOT fibers can also be considered as active devices (i.e., organic electrochemical transistor) and can reproduce neuromorphic features such as short-term and long-term memory effects[24]. These innovative devices, which are offering an interesting bottom-up framework for neuromorphic hardware engineering are also offering unique computing features that have been exploited in the context of reservoir computing. For instance, the iono-electronic coupling in dendrites was used for computing in the context of reservoir[25]. First experiments on simple 3-nodes problem such as Pavlov's dog learning[21,26] or binary classification[23] were also proposed and were pointing toward the benefit of using structural plasticity to realize actual computing functions. This concept marked a clear departure from conventional approaches, which were only relying on synaptic plasticity for learning. But it was still unclear to what extent the structural plasticity could contribute to improve computing efficiency at the network level. The next challenge toward the realization of fully evolvable neuromorphic networks with structural plasticity is to show that growth of dendritic connections can reproduce neurons' genesis on more complex networks and to demonstrate that the resulting networks are optimal topologies for computing. This would offer new perspectives from both materials and devices engineering, but more fundamentally to the progresses toward brain-inspired computing systems.

## Results

### Hebbian-like structural plasticity with time correlation

Bipolar AC electropolymerization of PEDOT:PSS fibers is realized in water-based electrolyte containing 10 mM of 3,4-ethylenedioxythiophene (EDOT) and 10 mM of benzoquinone as oxidizing and reducing species, respectively. The electrolyte also contains 1 mM of NaPSS as a supporting salt. Dendritic growth of PEDOT:PSS conducting fibers is obtained through application of bipolar AC voltages in between two gold wires immersed in the electrolyte. Such electropolymerization process has been reported to depend strongly on electrical parameters of the AC input signals such as peak-to-peak amplitude and frequency[23]. These dependencies were correlated to the voltage potential oxidation threshold of EDOT monomers into polymers and to the dynamics of charged species in the electrolyte through the influence of drift and diffusion[27]. The structural aspect and growth dynamics of dendritic fibers are reminiscent of neurons genesis during development of biological neural networks (see Supplementary Video 1). During biological network genesis, multiple parameters influence the topological organization of dendritic connections in between pre- and post-neurons layers. The coupling of various mechanisms is the key to defining the optimal, or at least efficient, topology for computing. While the details of biological processes are still not completely understood, we propose using a simplified description of these processes and turning them into a hardware implementation.

The first parameter is the geometrical organization of cells, which will determine how close two cells are to each other. Cells being closer will inevitably have a higher probability of connecting together. A second parameter is the correlated activity in between cells that influence their probability of connection. This aspect could be realized through diffusion of chemical molecules and ions in between pre- and post-neurons favoring directional dendritic projection and leading to a specific interconnection pattern[14]. This idea was the central point of Hebb's theory and was extensively used for deriving synaptic plasticity rules[28]. In a first step, we propose to reproduce with conducting dendritic fibers the implementation of this second parameter through an adaptation of Hebb's principle to structural plasticity rules. The implementation of the first parameter will be discussed later in this article with two-dimensional topologies. A schematic overview of the concept is presented in Fig. 1.

PEDOT:PSS dendritic fibers growth results from the electrical potential at the interface between the Au electrode and the electrolyte. In a simple two-wire setup, where each electrode is respectively associated with the pre- and post-neuron, the effective voltage driving electropolymerization is the difference in between $V_{pre}$ and $V_{post}$, the electrical voltage of each electrode. Pre- and post-neuron activities are emulated in our setup as bipolar pulses of width $T_W = 2.5$ms, amplitude $V_{pre}=V_{post}=V_p=5$V, and mean frequency F=$f_{pre}$=$f_{post}$. A first convenient implementation of correlated activity in between two neurons is to consider spike-timing difference in between the pre- and post-cells. Short time difference in spike timing is associated with a strong correlation of activity while large time difference is associated to low correlation. Bipolar pulses can be conveniently designed to reproduce this correlation through the effective voltage overlap resulting from the potential difference $V_{pre} - V_{post}$ (Fig. 2a). Pulses overlapping results in overpotential leading to electropolymerization. The larger the correlation, the larger the duration of the overpotential (see supplementary information, Fig. 1). Note that no overlap in between pulses didn't lead to any dendritic growth in the time scale of our experiment. Figure 2b presents the evolution of the dendritic network for different correlations of activity associated with time difference $\Delta T$ from 0 ms (strong correlation) to 2 ms (low correlation). Multiple pulses at mean frequency $f_{pre}$ =$f_{post}$ were applied continuously in between the two nodes and lead to dendritic growth (see Supplementary Video 1). Several mean frequency values of 20 Hz, 80 Hz, and 130 Hz were tested. Note that the bipolar pulses remain the same for the different frequencies (width and amplitude remain constant), only the time difference in between two pulses is modified. Higher activity correlation (i.e. decrease of $\Delta T$ and increase of overpotential duration) clearly led to denser dendritic trees, creating more connections in between PEDOT:PSS fibers. This is supported by the measurement of the electrical conductance after pre- and post-dendrites connect each other. Higher correlation of activity between neurons leads to more synaptic connections established in between the dendritic branches, and consequently higher synaptic weight (i.e. higher conductance). This effect is in agreement with previous report[23] that associated the number of

branches with the voltage amplitude and frequency of the bipolar pulses in AC electropolymerization experiments. In our experiment, increasing the correlation in between pre- and post-pulses lead to a higher mean voltage experienced by the dendrites and a more branchy structure. On the contrary, structures with fewer synaptic connections and correspondingly lower conductance are observed at lower signal correlation. Note that this conductance can be further modulated by short- and long-term plasticity processes (STP and LTP) as in refs. 24,29.

Additionally, final conductance shows some moderate dependency on pulses frequency, except for the 20 Hz case. This latter effect could be explained by the reduced relaxation time between two electropolymerization events. In between pre- and post- pulses overlapping, the electrodes are grounded and charged species in the vicinity of the electrodes diffuse back into the electrolyte. Thus, at low frequency (20 Hz), this phenomenon results in fewer species available for electropolymerization, hence fewer brunches and lower conductance. Finally, the growth rate of the dendritic branches does not depend on time correlation but mostly on pulse frequency. By considering the variability of the dendritic growth process, a quasi-linear relationship in between growth rate and frequency could be considered. This would imply that each electropolymerization event (i.e. pulse overlapping) would contribute to longitudinal growth while $\Delta T$ affects the number of branches (i.e. the final conductance).

## Associative learning with structural plasticity

We illustrate the interest of such time-based structural plasticity for defining network topology with Pavlov's dog associative learning example. In this demonstration of associative memory, the objective is to show how correlated activity in between multiple nodes can induce selective neuron's recruitment. In Pavlov's conditioning experiment, the correlation of neuron's activity between the sight of food and salivation is first established. Note that this correlation is the result of association between food and saliva in the dog's physiology during feeding. We are limiting our demonstration to the essential association step and not reproducing the full conditioning task that would require

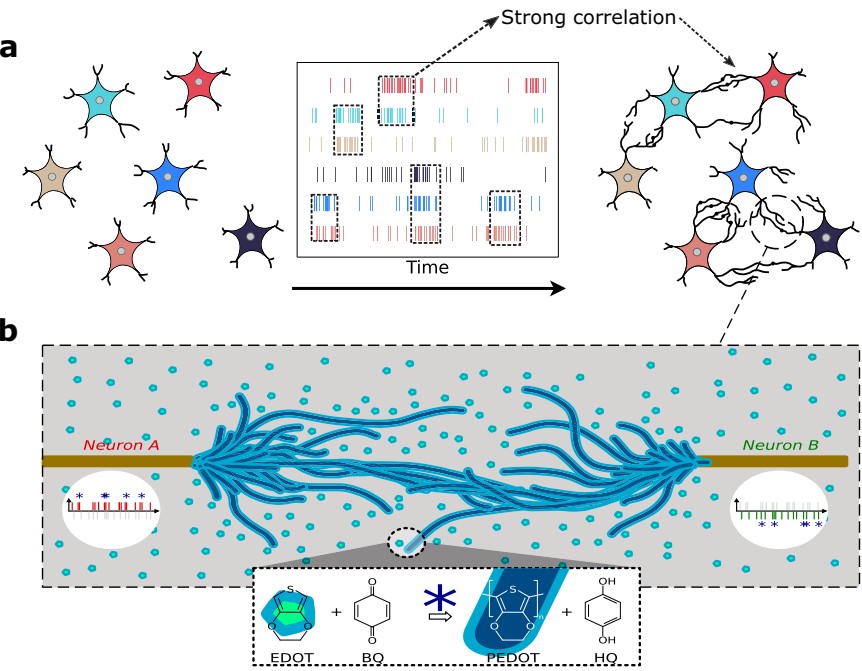

**Fig. 1 | Structural plasticity emulation with PEDOT:PSS dendritic fibers. a** As the activity correlates between every pair of neurons, the dendrites between them grows, creating a physical topology of artificial neural networks. **b** Dendritic electropolymerization is obtained by applying AC electrical signal in between conducting Au wires. Electrical potential at each node drives oxidation of EDOT molecules into PEDOT and reduction of benzoquinone (BQ) into hydroquinone (HQ). A pulse of voltage on each terminal represents node activity that can correlate to create overpotential (blue star), leading to electropolymerization.

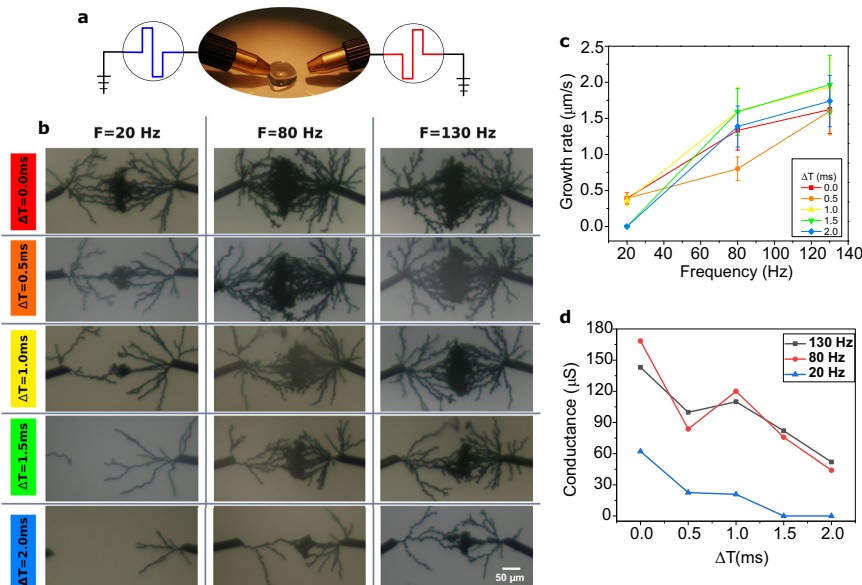

**Fig. 2 | Dendrites electropolymerization induced by spike timing modulation.**
**a** Setup for dendrites electropolymerization induced by spike-timing activity of applied signals. Bipolar pulses with opposite polarity were applied from both electrodes to emulate activity of neurons with time difference in between both of $\Delta T$. **b** Optical microscope images of completed dendritic branches obtained when $\Delta T$ time shift increases. Bipolar pulses are repeated at frequency $f_{pre} = f_{post} \in \{20$ Hz, 80 Hz and 130 Hz}. No connections occurred at $\Delta T = 1.5$ ms and 2 ms for $F = 20$ Hz. **c** Longitudinal growth rate evaluation and **d** conductance evolution of dendritic connections synthesized at different $\Delta T$ values and at $f_{pre} = f_{post} \in \{20$ Hz, 80 Hz and 130 Hz}. Growth rate is evaluated by considering a gap distance of 240 μm and a completion time corresponding to the time when dendrites of both electrodes first connect to each other. Note that after bridging branches with each other, the dendritic growth saturates, and no further significant changes in the number of branches and branches diameter were observed (see supplementary, Figs. S11 and S12). Error bars in **c** are represented by the standard deviation expressed in percentage.

demonstration of memory extinction and memory recovery. These additional effects could be implemented via additional synaptic plasticity mechanisms that have been demonstrated in PEDOT:PSS-based dendritic fibers[24]. The full conditioning task would therefore result from a combination of structural plasticity (association) and synaptic plasticity (i.e. forgetting and recall). In our experiment, during the first phase of association, time-correlated signals between food (pre-neuron) and saliva (post-neuron) result in dendritic growth and the creation of synaptic junction, while the uncorrelated bell neuron doesn't connect (Fig. 3a). This correlation is physically represented through the overlaps of bipolar pulses from pre- and post-neurons. The difference of potential between the electrodes during correlated activity (in comparison with the potential difference between electrodes during uncorrelated activity) is maximal and eventually leads to electropolymerization and dendrites' growth. This strategy has been adopted in other works for reproducing biological spike timing dependent plasticity[30]. In the second phase, the bell's signal is now correlated to both sight of food and salivation. Additional dendritic branches grow in between correlated nodes, resulting in the recruitment of the bell signal to salivation (Fig. 3b). Note that the time to bridge the bell and food electrodes is twice larger than the time to initially bridge the food and saliva electrodes. This effect corresponds to the growth of dendrites occurring only from the bell electrode dendrite. This effect could substantially increase the completion time during network formation but could be mitigating by engineering the electrode shapes and dendritic projection as in ref. 31. This associative mechanism is expected to play an important role in learning in biology with unlabeled data[32,33]. The association between neurons is based on correlated activities, and it occurs when neurons are recruiting other neurons to establish functional neuronal circuits. In the context of learning, such self-organized neural topologies could explain the ability of biological networks to generalize on very few examples, while fully-connected ANNs require extensive training examples to learn through synaptic plasticity. The network topology could define a general memory content (i.e, coarse-grain tuning) and synaptic plasticity can be used as a specialization on few examples (fine-grain tuning) leading to fast learning process. When synaptic plasticity alone is used, fine-grain tuning requires much more examples for learning. From a higher level, combining structural plasticity with fine-grain synaptic plasticity tuning could be a corner stone to solve the binding problem[9], which try to reconcile symbolic approaches (i.e., hierarchical structure in the data representation) with the neural representation (i.e., elementary symbol representation on each node of the network). Structural plasticity could be seen as the general architecture of the network bearing symbolic rules expression, while synaptic plasticity could be associated to the learning of specific representations.

## Generalization of structural plasticity with Poisson-sampled spike trains

A more generalized formulation of the structural plasticity through dendrites' electropolymerization is proposed in Fig. 4. Instead of considering pulse timing between pre- and post-neurons with fixed frequencies, we investigated the dendritic growth mechanism with Poisson-sampled spike trains. Now, the activity of a neuron is defined as the mean frequency of a Poisson distribution of bipolar pulses. This signal representation is often chosen to describe biological neuronal activity. The correlation of activity between two neurons $i$ and $j$ is proportional to the product between the two mean normalized frequencies:

$$\rho_{a_i, a_j} = \mathrm{corr}(a_i, a_j) \propto F_i \cdot F_j \tag{1}$$

This product describes the probability of having correlated pulse events between the two neurons when the pulse shape is fixed. The exact relationship between the correlation and this product is available in the Supplementary information, Fig. 6. Note that this expression is well adapted to describe Hebbian-like mechanisms, where the product

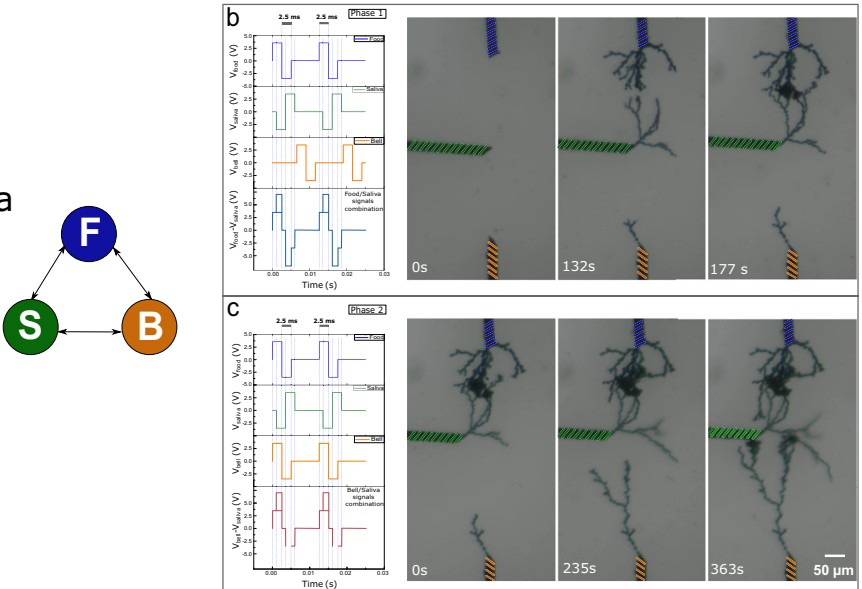

**Fig. 3 | Pavlovian learning promoted by means of spike-timing dependent activity electropolymerization conducted in two phases. a** Neuron-level schematic representing the growable connectivity between each pair of neurons, associated with food, saliva and bell. **b** Time-lapse of dendritic growth during 1st phase in which time correlation and corresponding overpotentiation took place only between "food" and "saliva" pulses (signifying that salivation naturally occurs only as response to the food stimuli). **c** Dendritic evolution with time during 2nd phase in which as a result of associative learning the correlated activity is as well introduced for "bell" and "saliva" signals leading to pairing of these nodes by polymer fibers.

of both pre-and post-neuron activity is required to induce wiring. In this more general situation, the same pulse overlapping mechanism as described previously is the driving force to induce dendritic growth in between two neurons, except pre- and post-neurons do not need to be specified (see Supplementary information, Fig. 5). Figure 4a shows an example of a 4-wires setup with one input neuron and three output neurons spiking with four different frequencies. The three output terminals are equidistant from the input terminal. Again, dendritic branching is obtained first in between input neuron and the most active output neuron. We performed multiple experiments with different $\rho_{a_i,a_j}$ to extract a representation of dendritic branching in Fig. 4b-c. Consistently with the previous results, higher $\rho_{a_i,a_j}$ results in larger conductance due to denser dendrites and more synaptic connections in between nodes.

These measurements were used to extract a dendritic growth model with overlapping pulses. In the model, a connection appears at time $t_{conn}$ when the Euclidean distance $d$ between a pair of neurons $i, j$ spatially positioned in two-dimensional space is equal to the integration of the overlaps between the two spike trains encoded as binary pulses $P_i(t)$ and $P_j(t)$:

$$\int_0^{t_{conn}} P_i(t) \cdot P_j(t)dt = d(n_i, n_j) \qquad (2)$$

Moreover, the conductance $G$ of the connection is dependent on the spike rate, which is modeled through synaptic traces $k_i$ and $k_j$. These traces are exponentially decaying with time constant $\tau$ and are incremented by a constant value $\Delta_k$ during a spike: $\tau \dot{k} = -k + \Delta_k \delta(t - t_{spike})$ with $\delta$ representing the Dirac delta function. This results in a similar integration where the conductance is equal to:

$$G = \int_0^{t_{conn}} P_i(t) \cdot P_j(t) \cdot k_i(t) \cdot k_j(t)dt \qquad (3)$$

Figure 4b, c presents the model's growth rate and conductance evolution with random Poisson-sampled spike trains of various frequencies as a function of $\rho_{a_i,a_j}$.

## Structural plasticity for classification task

Structural plasticity can be advantageously employed to solve the key issue of finding the sparse connectivity of neural networks. One of the important characteristics of biological networks' energy efficiency is their spatial sparsity (i.e., number of connections), which is difficult to obtain in conventional hardware implementations with dense fully-connected interconnections in between layers. We evaluate this property on a simple classification task presented in Fig. 5a. Surface electromyography (EMG) signals were recorded from the arm with three different movements of the hand corresponding to rock, paper and scissor in ref. 34. The eight temporal analog signals from each sensor are converted into 16 spike trains following the methodology described in ref. 35. The objective is to classify into three classes (i.e., rock, paper and scissor) the input spiking signals. The training dataset is composed of 300 multi-user samples and a testing set of 150 samples of the same users during a different session.

We tested two different network topologies where structural plasticity can be advantageously employed (Fig. 5c). Both networks are first wired using the structural plasticity model to create connections $w_{ij}$ in between the 16 input neurons $x_i(t) \in \mathbf{x}(t)$ and 3 output neurons $n_j(t) \in \mathbf{n}(t)$. For each sample in the dataset, the output neuron corresponding to the correct class spikes with a Poisson-distributed signal of high frequency, while the other output neurons are subject to a lower frequency activity (Fig. 5a), i.e., a soft one-hot encoding scheme. Correlated activity in between the 16 input and three output neurons is calculated on the entire training dataset by integrating over time the overlapping pulse duration. Heat map reveals that some nodes have higher correlations than others, see Supplementary information, Fig. 7. If we consider 16 input nodes equidistant from three output nodes, the cumulative effect of input/output signal correlation leads to gradual wiring in between nodes. This evolution over time is presented in Fig. 5b and is consistent with ref. 36. The dendrites propagate, and create connections with a given initial synaptic strength ($W_{init}$). Nodes with stronger correlation tend to connect earlier and define intermediate topologies with a gradual decrease in sparsity. Various levels of sparsity can be obtained in between the layers by stopping structural plasticity at different stages.

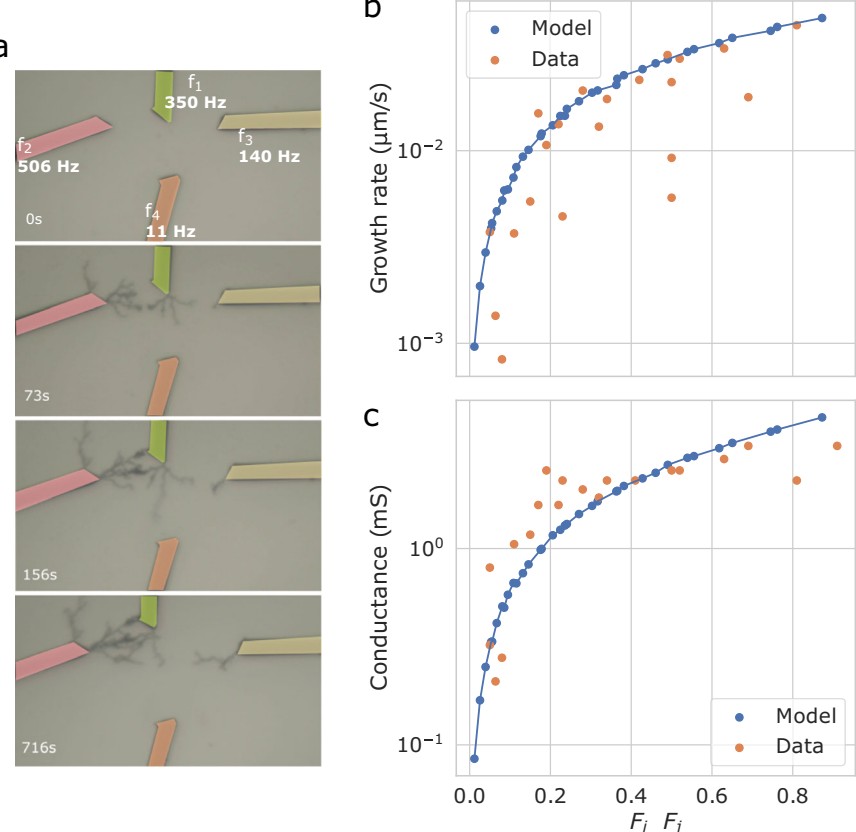

**Fig. 4 | Dendritic growth from data to model. a** Time-lapse of physical dendritic growth on four electrodes with unlike frequencies. Longitudinal growth rate (**b**) and conductance (**c**) of dendritic connections between wires stimulated by random Poisson-distributed pulses of various frequencies $F_i$ and $F_j$, compared with the dendritic growth model.

Once the connectivity is set, the synaptic strength $w_{ij}$ of the existing connections is adjusted through synaptic plasticity, which is dependent on the network topology and the objective performance of the task. During this second step, no structural plasticity occurs. The learning strategy is split into two cases:

Case 1: The initial weights $W_{init}$ are first binarized. Synaptic plasticity occurs in between the spike counts $\sum_t \mathbf{n}$ and a new atemporal output layer **o**. Synaptic weights in between $\sum_t \mathbf{n}$ and **o** are obtained through a linear support vector machine (SVM) classifier[37].

Case 2: The spikes are counted directly in the first layer $\sum_t \mathbf{x}$, and the sparse weight matrix $W$ is trained with an adaptive gradient method[38].

From literature, a fully connected non-spiking SVM classifier in between the input and output layer for this particular task can reach 85% recognition accuracy. Classification requires in this case $16 \cdot 3 = 48$ connections with floating point accuracy. Other work with reservoir-based architecture and high density of nodes performed in between 70% and 88%[35,39]. From Fig. 5d, we evaluate that > 80% recognition accuracy can be achieved with as few as 13 plastic connections in case 2. Interestingly, case 1 reaches an 81% peak performance with $9 + 13 = 22$ connections, showing that better connectivity matrices exist for this network configuration. We estimate the network accuracy on both the training and testing dataset in Fig. S13. The same maximum accuracy is observed on both datasets, thus ruling out the hypothesis of overfitting by increasing the number of connections. All cases of structural plasticity-backed networks demonstrate that activity-dependent topologies can outperform random or full connectivity, with a strong benefit on the number of total connections required for this classification task. An additional benefit that can be evidenced

from Fig. 5d is the better performances from synaptic weight with values $W_{init}$ in comparison to random initial weight, even with fully connected network. This aspect is highlighting the interest of defining initial weight values based on structural plasticity mechanism.

The network topology presented in Fig. 4b cannot be transposed directly to an actual hardware implementation of the single layer with grown connections. 3D implementation of the different nodes would advantageously enable parallel interconnections in between the input and output layers as proposed in ref. 31 but 2D integration remains the most straightforward mapping. Figure S9 presents a possible mapping of the single layer into a 2D problem that preserves the all-to-all connectivity. This integration of the different nodes also only requires growth of connections in between neighboring nodes, which is in line with the hardware constraint imposed by 2D dendritic growth (i.e. avoiding overlapping of dendrites).

### Structural plasticity for auto-encoder

Previous sections put the emphasis on the role of structural plasticity to obtain better performing topologies. In hardware implementations of structural plasticity, not only the activity in between nodes needs to be considered, but also the distance in between nodes, since neurons close to each other have initially a higher probability to connect together. The initial distribution of nodes poses a clear challenge, as this choice will strongly determine the final topology. We propose here to test the structural plasticity in the context of reservoir network. In the general reservoir framework, topologies are random. Nevertheless, it has been shown that some topologies performed better than others and strategies to find these topologies are still highly desired. We begin with a two-dimensional network of nodes organized into a square mesh topology (Fig. 6b). The objective of the task is to perform an

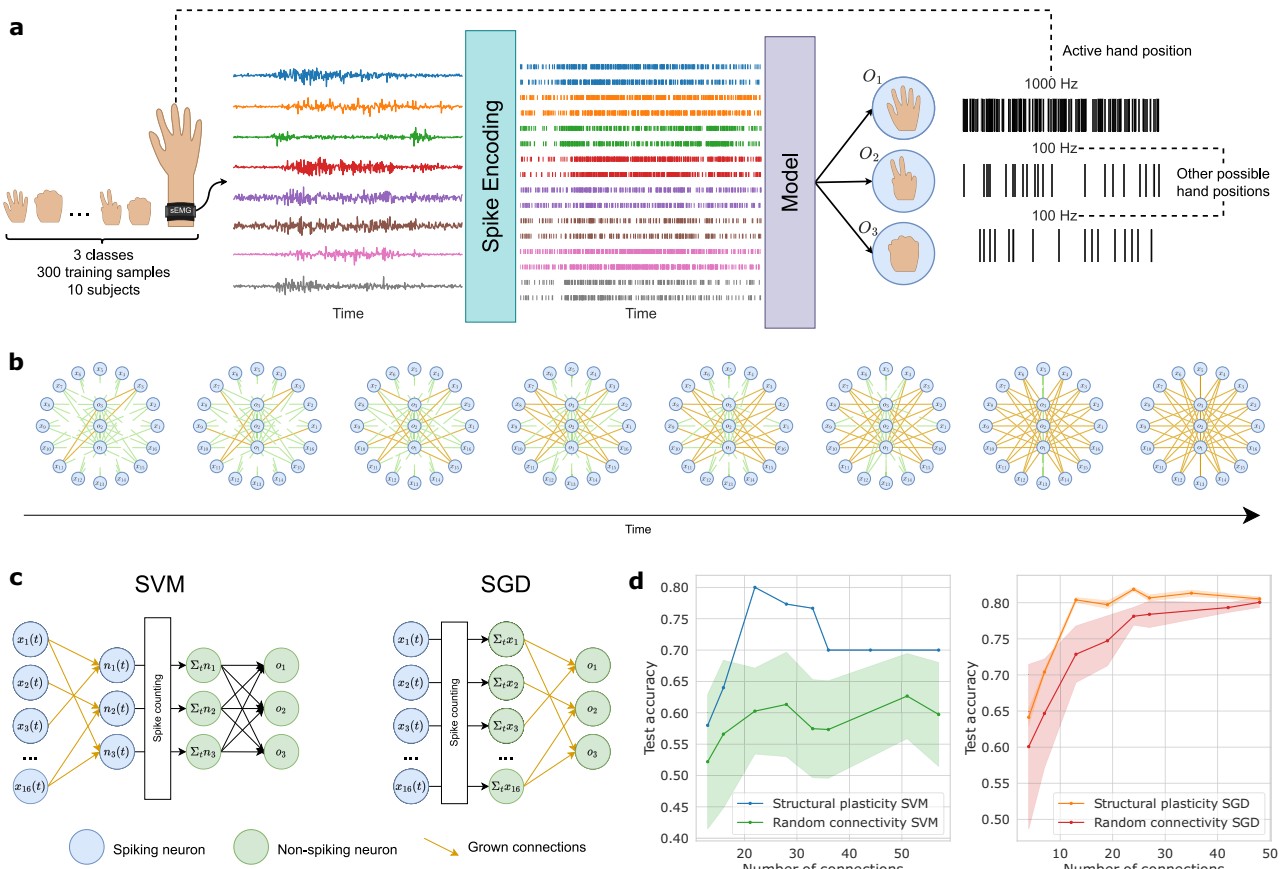

**Fig. 5 | Electromyography classification task. a** Schematic overview of the EMG classification task. Each input of the eight channels is first spike-encoded into 16 spike trains[35]. The resulting spike activity is passed through the structural plasticity model with three output neurons, creating the 16→3 connectivity matrix. The output neurons spikes at either 1000 or 100 Hz, associated with the correct and incorrect classes, respectively, of the currently presented label. **b** Resulting topologies of the dendritic growth stopped at various points in time. Partial connections are indicated in green, and fully connected nodes are linked by a gold-colored link. The organization of the nodes is schematic and does not reflect the actual placement of electrodes. **c** Two different networks were tested with modeled structural plasticity data. In the first model, the spikes go through the sparse connectivity matrix with binary weights, and spike counts are accumulated before being classified with a SVM. In the second model, spikes are accumulated directly on the 16 inputs, and a sparse weighted tensor is trained with stochastic gradient descent. Weight training is done after the creation of the connectivity matrix using structural plasticity in all cases. **d** Test accuracy of the EMG task as a function of the total number of connections. The two training methods are compared with a random topology with random initial weights, vs the structural plasticity-based topology. Ten different random topologies are generated and averaged for the random connectivity. Standard deviation is presented with a shaded area for each corresponding curve.

accurate reconstruction $\hat{x}(t)$ of a two channels analog signal $x(t)$ (Fig. 6a). Each input channel of the analog input signal $x(t)$ is randomly connected to 3 of the 25 reservoir LIF neurons, and their direct neighboring neurons. The input weights are fixed, and randomly sampled from a normal distribution $\eta(0, 0.25)$. The reconstructed signal $\hat{x}(t)$ is read from a linear fully-connected read-out layer with two output neurons trained from the filtered spike train of the reservoir. The network learns to encode the input signal with a compressed spike-based representation, following the work of ref. 40. By doing so, the compressed representation can be transmitted with a lower bandwidth, or used in conjunction with a low-power neuromorphic device to solve a downstream task. The baseline for this task is presented in Fig. 6c. In the baseline experiment, connections in the reservoir are selected randomly. Then, synaptic plasticity is activated following the learning rule derived in ref. 40 and described in the supplementary materials. Augmenting the connection density in the reservoir increases the performance continuously until the fully connected network is reached. In the structural plasticity experiment, each neuron receives a random combination of the input signals and grow dendritic connections based on the correlation of activity in between nodes. Naturally, neurons closer to each other have a higher

initial probability of connecting. Figure 6b shows the gradual evolution of the network topology. Various levels of sparsity are evaluated for the reconstruction task using the structural plasticity rule with different parameters, followed by a synaptic plasticity rule for weight training. The performances show again a much quicker convergence toward the lowest mean squared error with as few as 50% of total connections.

The resulting topologies obtained in Fig. 6b are nevertheless not achievable in practice when mapping the reservoir network onto a 2D substrate since overlapping of dendrites cannot guarantee integrity of the connection in between two nodes (dendrites are not insulated). We conduct in Fig. S10 the same experiment as in Fig. 6 but with an additional constraint on the possibility of connection in between nodes. In this experiment, nodes can only connect to neighboring nodes, thus alleviating dendrites overlap. This choice is also limiting the projection space of the reservoir since not all the nodes are visible for a given input. Interestingly, Fig. S10 shows that the benefit of structural plasticity is preserved even with a much higher constraint imposed by the hardware mapping of dendrites. Future works should consider how scaling more complex networks on 2D hardware and how integrating additional constraints (i.e. small-world topologies,

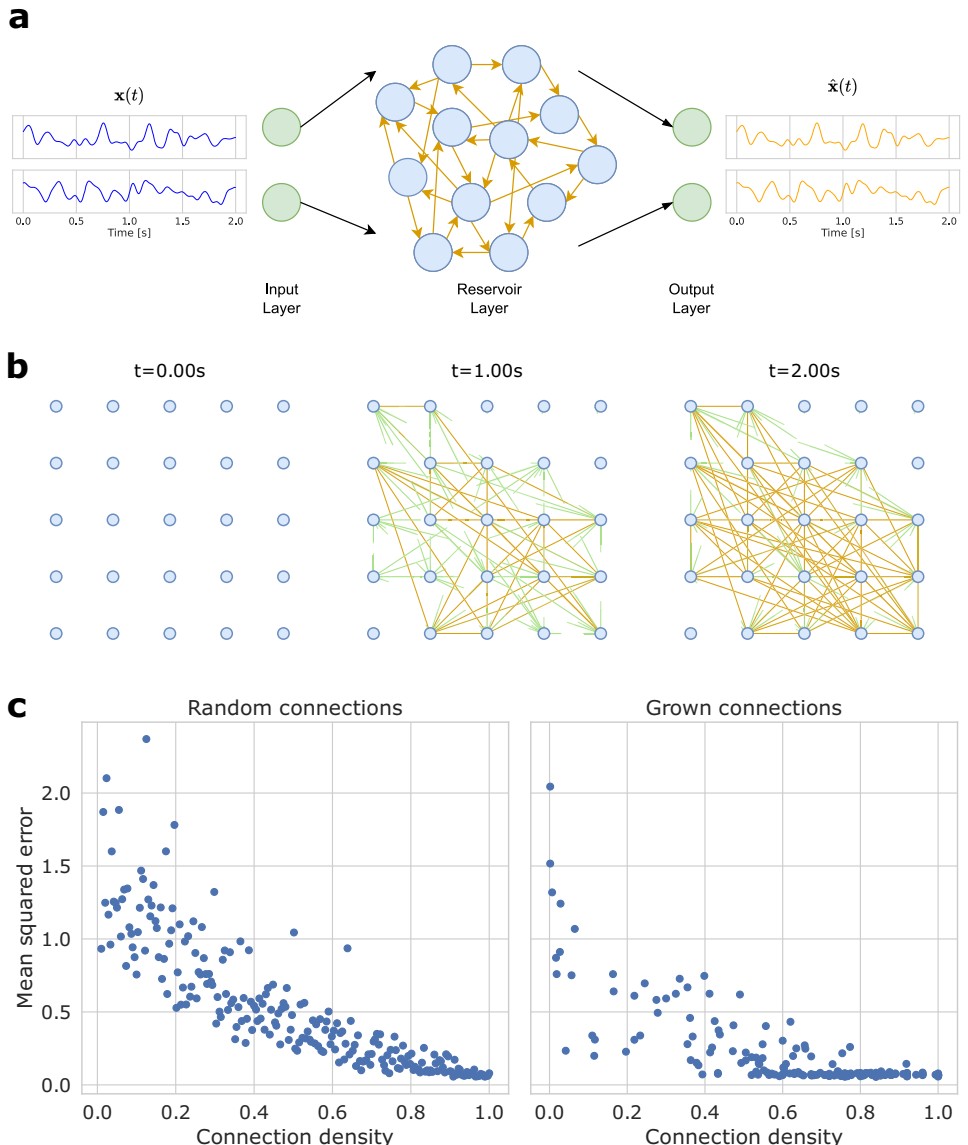

**Fig. 6 | Sparse unsupervised spike auto-encoding task. a** Schematic overview of the spike auto-encoding task. A random multi-channels input signal is connected by a linear input layer and fed to a pool of recurrent spiking neurons as currents. The spikes of the reservoir are filtered and decoded by an output layer to retrieve the original signal. The input layer is fixed and sparsely connected to the recurrent layer called the reservoir. The reservoir layer connectivity matrix is generated by either the structural plasticity method, or with random connectivity. **b** Two-dimensional lattice graph of spiking neurons at different point in time during the growth of connections with the structural plasticity model. The nodes' organization is simulated and does not depict the position of electrodes in a physical system. Each pair of neuron can grow a connection in-between. Green links represent partial connections, and gold-colored links are fully created connections. **c** Sparse unsupervised spike encoding error as a function of the connection density. The connection density represents the number of connections divided by the total possible number of connections. Self-connections are not included. The resulting mean squared error for encoding a random white-noise signal is reported. In the left plot, random connectivity matrices are generated for various target densities. In the right plot, the structural plasticity software model was used with different parameters (pulse duration and interneuron distance) with a fixed two-second duration as to create topologies with various densities.

etc.) In fact, the choice of nodes distribution on a 2D substrate is pre-defining the connectivity matrices that could be obtained after structural plasticity learning. This effect could mirror the combination of genetic and phenotypic evolution observed in biological networks.

## Discussion

In this work, we propose a novel structural plasticity rule associated with a hardware strategy for its implementation. Different tasks, which are representative of ANN and spiking neural network (SNN) applications, are demonstrating the benefit of structural plasticity for defining better topologies. Notably, we show that structural plasticity can define better-performing topologies with high level of sparsity in the

number of connections when implemented in various machine learning tasks. From a general hardware perspective, this is a clear benefit since it could minimize the memory requirement of learning systems. This aspect is beneficial from both a hardware complexity perspective (less memory is required) and from an energy consumption perspective, since memory access is still a major challenge for ANN and SNN implementations. From an algorithmic point of view, it is interesting to note that a simplified version of network genesis during development is able to reach a significantly higher performance than synaptic plasticity alone. This result suggests that there is a high potential of ANN and SNN optimization by integrating structural plasticity learning and to explore combination of synaptic plasticity rules with structural

ones. It is also highly beneficial to iterate only once over the synaptic plasticity training phase and avoid testing a huge set of random or semi-random topologies. An interesting issue that we didn't address in this paper is to consider the deployment of structural plasticity in multi-layer network. One of the key principles of Hebbian structural plasticity is that the presynaptic and postsynaptic node activity must be correlated for dendritic growth. In traditional ANNs, for connectivity to occur, there must be some kind of initial connectivity between the layers to have postsynaptic activity and enable dendritic growth. Otherwise, the input activity must be fed to all the layers to force some kind of initial activity. This is why in the EMG task the output neurons are stimulated with Poisson noise. In a multi-layer setting, it would be hard to decide which hidden neuron should fire and when, as they are not directly associated with the output labels. It would be possible to have spontaneous activity in the hidden layers to allow the creation of connections in a standard multi-layer ANN. This would be some departure from the Hebbian principle, and the effect on performance would need further study.

In terms of hardware implementations, engineering of conductive dendritic connections based on PEDOT electropolymerization is providing a new strategy for bio-inspired hardware design. Firstly, dendritic connections are realized into liquid environment and are opening the door to wetware engineering rather than conventional hardware (i.e., CMOS-based technologies). This could have a strong implication at the hardware level since it could offer the possibility to use multiple carriers for information representation such as ions and molecules and not being limited to standard electronic-based hardware design that are prone to high energy consumption. Secondly, this strategy could reinforce the analogy with biology for implementing multiple computing mechanisms that are hard to realize with conventional hardware, such as long time constant during learning or representation of various traces of information for learning implementation. While structural plasticity requires some amount of data to connect neurons together, the lower parameter count of the resulting network should enhance its ability to learn with fewer data samples. One-shot or few-shot synaptic plasticity techniques could go well with this approach to enable devices to be trained at low energy and latency cost on-the-fly. Combined with other biocompatible devices such as optical memristors[41], such fast-learning systems could be implemented and trained directly in vivo.

## Methods

### Materials and instrumentation
Dendrites formation was carried out in an aqueous electrolyte containing 1 mM of poly (sodium- 4- styrene sulfonate) (NaPSS), 10 mM of 3,4-ethylenedioxythiophene (EDOT) and 10 mM of 1,4- benzoquinone (BQ). All chemicals were used without any prior modification and purchased from Sigma Aldrich. 25 μm-diameter gold wires (purchased from GoodFellow, Cambridge, UK) were employed as electrodes. Gold (Au) wires were immersed into a 20 μl electrolyte drop and placed onto a parylene C covered glass substrate. In all the experiments, electrodes were equally lifted at a controlled height from the substrate. Each dendrite growth was conducted with a new pair of gold wires and daily prepared solutions. Recording of dendrites' formation process was captured with a VGA CCD color Camera (HITACHI Kokusai Electric Inc).

### Electrical characterization
Signals were generated in Waveform Generator/Fast Measurement Units (WGFMU), which were coupled with Agilent B1500A Semiconductor Device Analyzer and B2201A Switching Matrix.

## Data availability
All data needed to evaluate the conclusions in the paper are present in the paper and/or the Supplementary Materials.

## Code availability
All the codes are available from the authors upon request.

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

## Acknowledgements

The authors wish to thank the European Commission: H2020-EU.1.1.ERC project IONOS (# GA 773228). I.B. acknowledges the support of the Natural Sciences and Engineering Research Council of Canada (NSERC), funding reference number 559730. The authors thank the RENATECH network and the engineers from IEMN for their support.

## Author contributions

K.J., I.B., A.K., N.G., F.A., C.S. designed and performed the experiments. J.R., Y.C., S.P., F.A. supervised and administered the current study. F.A., D.D. provided funding to this project. F.A., I.B., K.J. wrote the initial draft of the manuscript. All authors provided critical feedback and helped shape the research, analysis, and manuscript.

## Competing interests

The authors declare no competing interests.
