## [Peer Review File · Nature Communications]

REVIEWER COMMENTS

Reviewer #1 (Remarks to the Author):

The authors are exploring the structural plasticity of polymer-based fiber networks grown by bipolar electropolymerization and they show the application of this effect for classical conditioning, electromyography classification and the realization of an autoencoder. The paper is interesting, well-written and technically sound. My main point of criticism though, is that I strongly question the novelty and impact of this work, which in my opinion does not justify publication in Nature Communications/ or after major revision.

To be more precise, the growth method and the structural plasticity that can be realized by the growth (network genesis) are not new and have been shown in several other publications, including work from the authors of this publication. Also the application of the structural plasticity for condition has been shown by others (compare e.g., <https://doi.org/10.1002/aelm.202100586>, SI). Furthermore, the demonstration of the autoencoder function or classification are only based on the plasticity model the authors developed in 2.3, but are not demonstrated in practice. Looking at the complexity of the fiber growth in a multielectrode array, I strongly doubt that the conclusion the authors draw on the better performance caused of the sparse network topology, are misleading and probably not achievable in practice. Or it should be demonstrated. The development of the plasticity model in 2.3. is in general interesting but also hand waving, looking at the work done previously by this group.

Major technical points:

- classical conditioning experiment: has been done by other groups as well, see comment above. Furthermore, you are not showing a classical conditioning experiment but only a small part of it, namely, the forwarding conditioning. Classical conditioning though contains several more features that you have not demonstrated: e.g., extinction of memory and spontaneous recovery.
- why does neither the fiber diameter nor the conductivity of the fiber network depend on the frequency? if I look at previous work from the same group but also others, I see this dependency
- The comparison of conductivity in 2.1. is difficult to understand as it is not clear when you stopped the growth process. In particular, the growth process continues in time and hence, the conductivity will increase if you grow further. So how did you define the time for the comparison of the conductivity, and it is then fair to compare networks grown at different frequencies.

- the evaluation of the performance for the classification task and the comparison to other system seems premature. Overall, the accuracy is fairly low, which in my opinion does not allow for a generalization of the finding that the sparse dendritic network is superior. It is only true for this specific task and encoding scheme, and showing it in simulation does not mean that it can be realized in practice looking at the complexity of the growth process.

Minor points:

- Language: several typos (e.g., oxydation), and inconsistent use of A.E. vs. B.E.
- not clear where the dataset for the electromyography is coming from? from [33]? Not clear how you did the spike encoding to make it fit to your plasticity model? Isn't there an arbitrary coupling factor connecting the frequencies?
- T_w value is not mentioned in the text.

Reviewer #2 (Remarks to the Author):

This paper addresses a long-standing problem in neuromorphic computing, specifically the optimization of network topology using Hebbian principles. The authors propose an organic network featuring structural plasticity based on PEDOT connections and emulate biological neural network development through the dendritic growth of PEDOT-based fibers via Au electrode and electrolyte. The results are both interesting and timely. However, several questions need to be addressed:

Q1: The authors should include dimension annotations in Figure 2b to correlate with the growth rate in Figure 2c and provide a definition and measurement criteria for longitudinal growth. Moreover, Figure 2b shows a negative correlation trend between dendritic branches' growth and ΔT , while Figure 2c reveals the maximum growth rate at $\Delta T=1.5\text{ms}$ for the 80Hz experimental results, creating a mismatch that requires clarification.

Q2: The authors should provide corresponding time units for the x-axis in Figures 3b and 3c (left panel), and a distance scale in the right panel. Additionally, it is unclear why, in the second phase, the growth speed of dendritic branches for Bell and Saliva is significantly slower than that of Food and Saliva in the first stage, despite the Food/Saliva and Bell/Saliva signals combinations having the same waveform (same frequency and ΔT).

Q3: The results in Figure 3 suggest that the formation of dendritic branches between two electrodes might affect the formation of dendritic branches with a third electrode. In the experiments shown in Figure 4a, should the mutual influence among multiple electrodes be considered, in addition to the frequency factor? Could this lead to adverse effects, particularly for models of interconnected neuronal networks?

Q4: In Figure 5b, a shift from partial to full inter-neuron connections is observed. It would be interesting to know if the process could be reversed—from full connection to partial connection. Additionally, in Figure 5d, a decrease in accuracy after the number of connections reaches 30 in the structural plasticity SVM model raises the question of whether this phenomenon could be explained by overfitting. If so, how can we determine the suitable number of connections and when to stop training?

Reviewer #3 (Remarks to the Author):

The manuscript by Janzakova et al describes a novel implementation of the previously reported in-situ growth of PEDOT:PSS dendritic fibers. The authors extensively explain how structural plasticity is achieved and then proceed to highlight three demonstrative applications where this can be used. Although the concept is not new, the implementation is excellent, the applications are very relevant and impactful, and the experimental achievements are state-of-the art and very extensive. I believe this work can be published in Nature Communications. Perhaps the authors can add an outlook of how these systems can be scaled up in the future, or whether there are physical limitations. In traditional ANN there can be many layers and many hidden nodes and neurons. How would that be envisioned in these systems? A simple perspective / discussion would help.

Reviewer # 1

The authors are exploring the structural plasticity of polymer-based fiber networks grown by bipolar electropolymerization and they show the application of this effect for classical conditioning, electromyography classification and the realization of an autoencoder. The paper is interesting, well-written and technically sound. My main point of criticism though, is that I strongly question the novelty and impact of this work, which in my opinion does not justify publication in Nature Communications/ or after major revision.

We thank the reviewer for their careful reading of our manuscript and comments. We propose below point-by-point answers to the reviewer's comments.

To be more precise, the growth method and the structural plasticity that can be realized by the growth (network genesis) are not new and have been shown in several other publications, including work from the authors of this publication. Also the application of the structural plasticity for condition has been shown by others (compare e.g., <https://doi.org/10.1002/aelm.202100586>, SI).

We agree with the reviewer that the concept of electropolymerization is not new and has been already described in major publications referenced in the main text of our manuscript (see [1–4]). In this work, we propose to use dendrites electropolymerization for computing and to show that structural plasticity is offering new perspectives from both a hardware and a software perspective.

[1] proposes to use a network of dendrites for reservoir computing. In this seminal work, the topology was chosen arbitrarily and the computing was obtained by taking advantage of the non-linear current-voltage response of the dendritic objects and by introducing artificial feedback to increase signal projection into the reservoir. In [3–5], the concept of structural plasticity (i.e. growth of material depending on stimuli) for computing appears for the first time with simple example such as Pavlov's dog learning or 2 inputs activity discrimination. These first contributions were all based on 2 inputs and 1 output nodes with dendritic growth occurring in between these 3 nodes depending on input/output signals combination. The dimension of the problem in these examples (i.e. 2 inputs 1 output) didn't allow to discuss the interest of structural plasticity at the network level.

We propose in this paper to go beyond these first propositions by (1) rationalizing the origin of structural plasticity with a Hebbian-like mechanism and (2) showing how structural plasticity can be used to find optimal topologies in networks. Such optimal topologies are found for two important neural network structures (single layer feed-forward network for classification and a reservoir recurrent network for signal reconstruction). In addition to the significant higher level of complexity in comparison with Pavlov's dog experiment and 2-signals discrimination, the major contribution of this work is to show that optimal topologies exist and can be found by structural plasticity. This key result has not been reported so far and could not be anticipated from 3-nodes problems such as Pavlov's learning or binary classification. This result constitutes consequently a strong foundation for the deployment of future neural networks. From a hardware perspective, this work shows that dendritic PEDOT fibers are bringing an additional key computing element (i.e. optimal topology identification) to existing neuromorphic hardware.

More broadly, this technic could be used in a variety of applications that go beyond neuromorphic computing. This result could be extended to a variety of computing problems such as graph theory, cellular automata or any optimization problem where nodes are connected into networks and topologies are not known *a priori*, but emerges from the network activity (e.g. network of communication with distributed nodes, electrical power management in networks and grids, transportation networks etc.). This contribution brings a significant level of novelty and justifies in our opinion publication in a journal with broad readership.

Modifications

1. We strengthened the paper positioning with respect to the literature in the introduction: “These innovative devices, which are offering an interesting bottom-up framework for neuromorphic hardware engineering are also offering unique computing features that have been exploited in the context of reservoir computing. [...] The next challenge toward the realization of fully evolvable neuromorphic networks with structural plasticity is to show that growth of dendritic connections can reproduce neurons’ genesis on more complex networks”.
2. We added one additional reference for the Pavlov’s dog demonstration [5] as pointed out by the reviewer.

Furthermore, the demonstration of the autoencoder function or classification are only based on the plasticity model the authors developed in 2.3, but are not demonstrated in practice. Looking at the complexity of the fiber growth in a multielectrode array, I strongly doubt that the conclusion the authors draw on the better performance caused of the sparse network topology, are misleading and probably not achievable in practice. Or it should be demonstrated.

We acknowledge that the scaling of the growth process, in practice, is an active and ongoing problem. We argue that validating the growth process at scale using software models is an important step toward the realization of fully-grown neuromorphic networks. There would be no point in scaling the network practically if the software models that we implemented did not show any improvement with grown connections. In this work, we provide empirical evidence, with two benchmarks, that structural plasticity done with PEDOT electropolymerization is a beneficial addition to neuromorphic networks. This result is the central point that bridges the software and hardware aspect of our paper.

The hardware demonstration of large-scale structural plasticity (i.e. at the network level) represents in our opinion the next step for this emerging topic. Several challenges could be identified in this direction, notably the question of multiple branches growth in between neighboring nodes (see also question Q3 from reviewer 2) and the practical mapping of abstract network onto physical substrate that could support the growth of complex networks (see also question from reviewer 3). We added in the revised version of the manuscript additional results that are pointing toward potential solutions to these challenges and strengthening the feasibility of hardware integration. Practically, the most straightforward integration on a physical substrate of dendritic growth is to

consider 2D network of nodes and in-plane growth of dendrites. We added in the supplementary material examples of 2D network on microelectrode array and dendritic growth obtained in between individual electrodes on arbitrary geometry. Branching neighboring nodes was realized in this example sequentially. This is supporting the feasibility of 2D integration of structural plasticity, but with the limitation of connecting neighboring nodes only (i.e. the connection in between two nodes separated by long distance (1) is compromising additional growth in its vicinity because of electric field screening and (2) requires high voltages that could result in water electrolysis and damaging of other dendrites). Additionally, such 2D network could be well adapted to simple network mapping. As an example, we propose in supplementary information a physical mapping of the classification task. In this 2D network, each node can connect only to its closest neighbor, which seems a realistic hardware constraint for dendritic growth. For the auto-encoding task, which requires a more complex network than a simple classifier in terms of connectivity matrix, we tested how this constraint of connecting only close neighboring nodes could impact the benefit of sparse network genesis with structural plasticity. Supplementary Fig. S10 shows that this hardware constraint still preserve the benefit of sparse network for this task. These different preliminary results are mostly included in the supplementary information to preserve the conciseness and readability of the paper, but are discussed in the main text of the revised version to support the feasibility of this work.

Modifications

1. Addition of 2D network examples on microelectrode array to support the hardware feasibility of complex 2D networks (SI) and discussion (main text).

2. Addition of hardware mapping strategy for classification task with close neighbor constraint (SI) and discussion (main text): “The network topology presented in Fig. 4b cannot be transposed directly for an actual hardware implementation of the single layer with grown connections. [...] This integration of the different nodes also only requires growth of connections in between neighboring nodes, which is in line with the hardware constraint imposed by 2D dendrite growth (i.e. avoiding overlapping of dendrites).”

3. Addition of close neighbor constraint on the auto-encoding task (SI) and discussion (SI + main text): “The resulting topologies obtained in Fig. 6b are nevertheless not achievable in practice when mapping the reservoir network onto a 2D substrate since overlapping of dendrites cannot guarantee integrity of the connection in between two nodes.”

The development of the plasticity model in 2.3. is in general interesting but also hand waving, looking at the work done previously by this group.

We previously proposed a phenomenological model describing the main parameters that impact the dendrites’ morphology [6]. This level of modeling is well adapted to describe a single dendrite and the growth dynamics but cannot be extended to network level simulations since the computing time would increase dramatically. Network level simulation always requires a trade-off in between complexity and speed. Here, we put the emphasis on the possibility to

integrate our model into a spiking neural network simulator in order to compute efficiently the network topologies obtained in Fig. 6. We believe that this model integrates the two main components of structural plasticity (i.e. growth rate and conductance) that are explaining the results of our paper. Future work could of course investigate in more detail the impact of variability in growth and dendritic projection cones on the network topologies obtained. We believe that these aspects need to be associated to a fix hardware integration and are of second order for the present paper.

Classical conditioning experiment: has been done by other groups as well, see comment above. Furthermore, you are not showing a classical conditioning experiment but only a small part of it, namely, the forwarding conditioning. Classical conditioning though contains several more features that you have not demonstrated: e.g., extinction of memory and spontaneous recovery.

Classical conditioning is indeed a complex task that we are not fully addressing in this paper. We rather used the Pavlov's dog experiment as a toy problem to show the key concept of associative learning. Associative learning is in essence a fundamental part of network genesis and a key step for classical conditioning. Forgetting and recovery are additional mechanisms to association that have been observed in psychology. We note that the mechanistic description of these two additional mechanisms are not completely identified in biology. We propose in this revised version to comment on this aspect and to propose a strategy that could reproduce these features. Namely, after association and growth of a connection, additional synaptic plasticity could be initiated and use to modify the strength of the dendritic branch. Interestingly, such an additional level of plasticity has been proposed in [7] and [8] but is not the objective of this paper.

Modifications

1. - Addition in the main text: “We are limiting our demonstration to the essential association step and not reproducing the full conditioning task [...]. The full conditioning task would therefore result from a combination of structural plasticity (association) and synaptic plasticity (i.e. forgetting and recall).”

Why does neither the fiber diameter nor the conductivity of the fiber network depend on the frequency? if I look at previous work from the same group but also others, I see this dependency

We report in our previous work [4] on the influence of the electrical parameters on the dendritic morphologies. Two main aspects were used to describe the dendritic morphology: (i) the branch diameter and (ii) the number of branches. We observed in this work that increasing the amplitude of a bipolar square shape voltage led to more branches. This effect was understood as higher voltage promotes more electropolymerization sites along the dendrites and result in a higher probability of nucleation sites. Increasing the frequency was associated to thinner branches and less numerous branches. This effect was understood in the light of response time of the charged species to an electric field in the electrolyte that present a cut-off frequency (i.e. higher frequency leading to less

species accumulated at an electrode). A second effect was identified in [6] and associated to the wire-like growth. At high frequency, charged species doesn't have time to diffuse in between two polarities and are trapped in the middle of the electric field lines (after all species present locally have been consumed).

In the present paper, the signals used to grow the dendrites present some significant differences. The bipolar pulse duration during which oxidation/reduction can occur is always 2,5 + 2,5 ms (two successive positive and negative polarities, see Fig. S1 in the supplementary information). In between each event, the electrodes are grounded. The effective time for dendrites to grow is constant through all the experiments, even if the mean frequency is changed (i.e. distance between two bipolar pulses). This could explain why the fiber diameter remains constant in between all experiments. We have now to explain why (i) increasing ΔT (the shift in between pre and post signals) and (ii) the mean frequency of the signals is increasing the number of branches. (i) When ΔT is reduced, the mean voltage seen by EDOT molecules increases (see example in Fig. S1, supplementary information). This increase in mean voltage is in agreement with more branches observed in our previous work. In addition, a decrease in ΔT leads to an increase in the duration of the effective voltage overlap and, therefore, more time for species to be accumulated and oxidized on the electrode, which eventually results in more branches. This observation also aligns with findings in the previous work. (ii) Increasing the mean frequency of the signals lead to a decrease of relaxation time in between redox events (i.e. time during which the system is grounded and charge species are repealed from the electrodes). This decrease in relaxation could lead to an accumulation of charged species in the vicinity of the electrodes/dendrites, in agreement with an increase of the number of branches.

Modifications

1. Addition in the main text: “Note that the bipolar pulses remain the same for the different frequencies (width and amplitude remain constant), only the time distance in between two pulses is modified. [...] This effect is in agreement with previous report [4] that associated the number of branches with the voltage amplitude and frequency of the bipolar pulses in AC electropolymerization experiments. [...]. Additionally, final conductance shows some moderate dependency on pulses frequency, except for the 20 Hz case. This later effect could be explained by the reduce relaxation time in between two electropolymerization events. In between pre and post pulses overlapping, the electrodes are grounded and charged species in vicinity of the electrodes diffuse back into the electrolyte, thus leading to less potential for electropolymerization.”

The comparison of conductivity in 2.1. is difficult to understand as it is not clear when you stopped the growth process. In particular, the growth process continues in time and hence, the conductivity will increase if you grow further. So how did you define the time for the comparison of the conductivity, and it is then fair to compare networks grown at different frequencies.

The growth process of each individual dendrite was carried out until completion time, the moment when the spacing between the dendrites from both

Au wires became zero. After that, the bipolar pulses were applied for another 30 seconds and then stopped. During this 30 s period, the dendritic structure remains unchanged, and further bipolar pulses application results neither in the development of new branches from the Au wires nor in a change in the branches' diameter. Minimal modifications appear only at the interjunction between branches from both electrodes due to their bridging. Such growth behavior could be explained by a redistribution of the voltage potential drop in the system electrodes/dendrites/electrolyte when the dendrites connect each other.

The minimal modifications that occurred in those 30 seconds are only related to the number of bridged branches. This period allows dendritic branches from both Au wires to fully interconnect. The final value of conductance is indeed affected by the number of joined branches. However, such a choice of protocol is justified by the fact that it is difficult to visually assess the level of interconnection. Moreover, it would be incorrect to compare a structure that has, for example, only a few interconnected branches out of five possible with a second completely interconnected structure. Due to this, it was decided to give the dendritic branches a certain time- 30 seconds to fully interconnect with each other. Thus, once dendritic branches are fully interconnected, it is more reasonable to measure and compare the conductance values of different dendritic structures.

To help visualize the growth process and assess the level of interconnections, one can consider the evolution of the effective dendrite area (dendrites are projected on a 2D image and analyzed using ImageJ software). For example, in our previous work "Analog programming of conducting-polymer dendritic interconnections and control of their morphology" 1, Fig. 2 f shows the changes of the 2D projected area with time for different dendrites grown at various V_p ($V_{off} = 0V$, $f = 80Hz$, $dc = 50\%$). We report on Fig. S11 in supplementary information, an example of the projected area changes over time for dendrites obtained at $V_p = 5.5 V$ ($V_{off} = 0V$, $f = 80Hz$, $dc = 50\%$). Here, the AC signal, after the completion time, continued for a time that was a little more than 30 seconds.

As seen in Supplementary Fig. S12, the projected area linearly increases (sub-linear regime), indicating the growth of dendritic branches from both electrodes before the completion time (images depicting growth evolution are shown at the bottom of the graph). The first image on the top of the graph shows the dendrites at the completion time when the spacing between branches becomes zero. From the completion time (233 s) and further, the projected area grew but not so intensively, and from 263 s, it reached a plateau. No significant morphological changes in the structure of the dendrites occurred during these additional 30 s. At that time, PEDOT:PSS grew slightly but only in the interjunction area, in the middle of the structure, indicating the bridging of branches from both sides (images on top of the graph, Supplementary Figure S12). The plateau on the graph indicates a saturation regime in which dendrites have the maximum number of joints between branches. Therefore, these 30 seconds are an appropriate period to allow the branches to bridge and then compare the measured conductance values.

The same growth behavior was observed for dendrites grown according to spike-time and spike-rate dependent modulations. For example, Supplementary Fig. S11 shows the changes in 2D effective area of dendrites grown at combinations of signals (bipolar pulses) with various frequencies F_i and F_j (see Fig.

4). We can see that after a sublinear regime (the dendritic growth), each experiment saturates to a fixed density (growth completion), indicating the full connection of branches.

Thus, the above growth protocol was applied to all types of dendrites, allowing them to fully interconnect. This also applies to high frequency and low frequency dendrites (section 2.1), so it is fair to compare their conductance values.

Modifications

1. Addition in main text “Note that after bridging branches with each other, the dendritic growth saturates and no further changes in the number of branches and branches diameter were observed (see supplementary, Fig. S11 and Fig. S12).”
2. Supplementary Fig. S11

Figure S11: Dendritic growth evaluated through image analysis for different growth frequency (see Fig. 4). Density corresponds to the 2D projection of the dendrites. The surface was analyzed with ImageJ software.

3. Supplementary Fig. S12

Figure S12: Projected area over time for dendrites grown at $V_p = 5$ V ($V_{off} = 0$ V, $f = 80$ Hz, $dc = 50\%$) with microscopic images showing dendrites' changes at different growth times.

the evaluation of the performance for the classification task and the comparison to other system seems premature. Overall, the accuracy is fairly low, which in my opinion does not allow for a generalization of the finding that the sparse dendritic network is superior. It is only true for this specific task and encoding scheme, and showing it in simulation does not mean that it can be realized in practice looking at the complexity of the growth process.

The accuracy is a metric that is highly dependent on the noise and reliability of the underlying data. While our results could be seen as not objectively high, they are in concordance with other state of the art and purely software-based implementations. To the best of our knowledge, the state-of-the-art result for this task are 88% accuracy with an optimized reservoir of 320 neurons [9]. Standard reservoir approach with the same 320 neurons has an accuracy of 77% [10]. Our 81% performances are in good match with digital and analog/digital approaches for this classification task. More importantly, our network does so using a fraction of the connection density of prior works, which also makes the comparison not as direct if low energy consumption is part of the objective of the network.

In this paper, we propose two tasks with a moderate level of complexity (feed-forward network classification and recurrent network for auto-encoding). Both are pointing toward the benefit of structural plasticity to find optimal topologies. Extending this work to more complex networks (i.e. deep networks) and finding strategies to map the structural plasticity algorithm to more complex network in order to generalize the interest of structural plasticity is definitely the next step for this project. We provide in the answer to reviewer #3 an extended discussion about these challenges and possible solutions.

Regarding the complexity of the growth process and the possible hardware mapping of the classification task, we referred to the second question raised by reviewer #1.

Language: several typos (e.g., oxydation), and inconsistent use of A.E. vs. B.E.

We have edited the manuscript to fix these typos and inconsistencies.

*Not clear where the dataset for the electromyography is coming from? from [33]?
Not clear how you did the spike encoding to make it fit to your plasticity model?
Isn't there an arbitrary coupling factor connecting the frequencies?*

The dataset comes from [11]. We clarified this point in the main text. The spike conversion was done in [9]. From the EMG data of 2000 ms, 600 ms in the beginning and at the end was trimmed. Spike encoding involves converting an analog signal into spikes. It detects amplitude changes over time and generates spikes when the difference exceeds a threshold, we have used a threshold of 0.5 in our treatment. The threshold value can change the mean spiking rate of the signal, in our present case the frequency of the encoded signal is 100 Hz to 200 Hz. The time series was constructed with a step size of 0.1 ms, thus the total 800 ms (2000-600- 600 ms after cropping) data was represented in 8000 points. The specific time intervals denoted by τ trigger these spike events. At each spike event, a bipolar pulse is produced with amplitude A from $(\tau - w)$ to τ and $-A$ from τ to $(\tau + w)$, with w representing a fixed duration of 0.4 ms and A represents the voltage of the signal. Similarly, the signals generated at output nodes carrying specific frequencies are generated by generating Poisson's train of a particular frequency. The output frequency was 1000 Hz for the correct label and 100 Hz for the incorrect label as depicted in the figure based on soft one-hot learning. Similarly, the bipolar pulses are generated at the spike points for the output series. The correlated activities between input and output time series are obtained. Based on the electropolymerization conditions, each correlated pulse would increase the dendritic length. Thus, the higher would be the correlation, the shorter the time taken to connect the input-output dendrite. In our present work, we have not evaluated the absolute time of compilation. However, the correlation map clearly defines the order of occurrence of connections, which is used in our structural plasticity rules.

T_w value is not mentioned in the text.

We have added this value in the main text and supplementary information.

Reviewer # 2

This paper addresses a long-standing problem in neuromorphic computing, specifically the optimization of network topology using Hebbian principles. The authors propose an organic network featuring structural plasticity based on PEDOT connections and emulate biological neural network development through the dendritic growth of PEDOT-based fibers via Au electrode and electrolyte. The results are both interesting and timely. However, several questions need to be addressed:

We thank the reviewer for their careful reading of the manuscript and comments. We propose below point-by-point answers to the reviewer's comments.

Q1: The authors should include dimension annotations in Figure 2b to correlate with the growth rate in Figure 2c and provide a definition and measurement criteria for longitudinal growth. Moreover, Figure 2b shows a negative correlation trend between dendritic branches' growth and ΔT , while Figure 2c reveals the maximum growth rate at $\Delta T=1.5\text{ms}$ for the 80Hz experimental results, creating a mismatch that requires clarification.

The longitudinal growth rate was calculated here as the ratio of the distance between electrodes with the dendrites' completion time (t_c) in μm per s. The distance between electrodes was always constant and was equal to $240\ \mu\text{m}$. The completion time was determined for each individual dendritic structure by visual examination of its growth on the recorded video, where the first moment of dendritic branches joining from both Au wires (when the distance between the dendritic branches from both Au wires becomes 0) was considered as the completion time. The completion time for Fig. 2b are provided in table S1.

		Frequency		
		130 Hz	80 Hz	20 Hz
ΔT	0 ms	148 s	180 s	612 s
	0.5 ms	150 s	300 s	614 s
	1.0 ms	124 s	150 s	667 s
	1.5 ms	122 s	151 s	0 s
	2.0 ms	138 s	173 s	0 s

Table S1: Completion time extracted for the different experiments in main manuscript Fig. 2.

The maximum growth rate for 80 Hz was observed at $\Delta T = 1.0\ \text{ms}$ and $\Delta T = 1.5\ \text{ms}$. and the minimum was observed at $\Delta T = 0.5\ \text{ms}$ (see Supplementary materials Table 1). Nevertheless, evolution of growth rate with ΔT is subject to variation in completion time from experiment to experiment. Such a variation has already been shown in Supplementary materials of “Analog programming of conducting-polymer dendritic interconnections and control of their morphology” article (Fig. R1).

Figure R1: Relationship between completion time of dendrites and applied various V_p ($V_{\text{off}} = 0\ \text{V}$, $f = 80\ \text{Hz}$, $dc = 50\%$). Adapted from [4].

Fig. R1 displays the dendrites' completion time grown at different VP and

Figure 2c: Longitudinal growth rate evaluation of dendritic connections synthesized at different ΔT values and at $f_{pre} = f_{post} \in \{20 \text{ Hz}, 80 \text{ Hz and } 130 \text{ Hz}\}$ with error bars

fixed frequency (f), voltage offset (V_{off}) and duty cycle (dc) parameters. Additionally, completion times of 8 different dendrites developed at similar conditions ($f = 80 \text{ Hz}$, $V_P = 5 \text{ V}$, $V_{off} = 0 \text{ V}$, $dc = 50\%$) is presented. All of these 8 dendritic connections showed different completion times, representing variability in growth rates even under the same electropolymerization conditions (electrical parameters). These data were used to calculate the error bar-standard deviation, which was 20.48% from average value.

Considering that Spike Timing modulation experiments were also carried out at similar electropolymerization conditions (electrolyte content, the same setup, bipolar pulses' parameters: 80 Hz, 5 Vp and $V_{off}=0 \text{ V}$), we applied this variability for ΔT experiments.

Error bars were integrated into the plot in Fig. 2c. Within the same frequency value, the error bars of different ΔT points overlap. This suggests that statistically speaking, these data points are not significantly different from each other within the margin of error represented by the error bars. In other words, the observed variations are likely within the expected range of random fluctuations, and there is no significant distinctions between these data points. Only the point of $\Delta T = 1.5 \text{ ms}$ at 80 Hz is outside this overlap, which can be related to the operator error. Values of $\Delta T = 1.5 \text{ ms}$ and $\Delta T = 2 \text{ ms}$ at 20 Hz are outside this overlap trend because no dendritic connections were established at these parameters and the growth rate at these points is considered to be 0 $\mu\text{m/s}$.

Thus, the absence of a linear trend in which the reduced number of branches at various ΔT correlates with growth rate is only the result of general data variation for completion time.

Modifications

1. Addition in main text: “Growth rate is evaluated by considering a gap distance of 240 μm and a completion time corresponding to the time when dendrites of both electrodes first connect to each other. Note that after bridging branches with each other, the dendritic growth saturates, and no further significant changes in the number of branches and branches

diameter were observed (see supplementary, Fig. S11 and S12).

2. Error bars in Fig. 2c and addition in the text on the competition between voltage and frequency: “Additionally, final conductance shows some moderate dependency on pulse frequency, except for the 20 Hz case. [...] This would imply that each electropolymerization event (i.e. pulse overlapping) would result in the same longitudinal growth while ΔT affects the number of branches (i.e. final conductance).”
3. Additional figure/table in supplementary information.

Q2: The authors should provide corresponding time units for the x-axis in Figures 3b and 3c (left panel), and a distance scale in the right panel. Additionally, it is unclear why, in the second phase, the growth speed of dendritic branches for Bell and Saliva is significantly slower than that of Food and Saliva in the first stage, despite the Food/Saliva and Bell/Saliva signals combinations having the same waveform (same frequency and ΔT).

There is indeed a ratio of 2 in the growth rate in between phase 1 and phase 2 in Fig. 3. This effect comes from the absence of longitudinal growth from “saliva” electrodes. Only longitudinal growth from “bell” electrode is occurring. Since growth occurs from both wire in phase 1, the speed is reduced by approximately 2 in the second phase when growth occurs only from one wire. We extend the discussion on this important aspect in the next question.

Q3: The results in Figure 3 suggest that the formation of dendritic branches between two electrodes might affect the formation of dendritic branches with a third electrode. In the experiments shown in Figure 4a, should the mutual influence among multiple electrodes be considered, in addition to the frequency factor? Could this lead to adverse effects, particularly for models of interconnected neuronal networks?

As we pointed in our response to Q1, the dynamic of dendritic growth is strongly dependent on the bridging between two nodes. Once a dendrite bridges two nodes, no further growth is observed in between these two nodes since electric potential drop occurs along the electrode/dendrite/electrode (i.e. the potential drop at the dendrite/electrolyte interface is not strong enough anymore). In the case of a multi-nodes setup, dendritic growth will still occur in between nodes that are not connected. The impact of additional dendrites will mostly impact the completion time, as observed in Fig. 3, where dendritic growth in phase 2 takes twice more time to bridge the nodes. This adverse effect is mostly due to the geometric organization of the dendrites that are growing along the electric field lines that are more intense at the apex of electrodes/dendrites. In order to prevent this effect, electrodes with multiple edges could be designed to favor growth in multiple directions. This effect is not included in the simulation of the classification task or auto-encoding and would require a more in depth analysis in future work. We believe that mitigating this effect should be done by engineering strategies associated to the physical implementation (such as electrodes with multiple edges or pre-oriented apex in between nodes), but cannot be described with enough detail in the setup we are using for this demonstration.

The second adverse effect that we could expect from multiple dendrites is the screening of electric field by dendrites growing in multiple directions, and mostly overlapping each others. We propose in the revised version of the manuscript an extended discussion about the possible hardware implementation of structural plasticity. This point is detailed in the answer to reviewer#1. Notably, we propose to leverage the impact of multiple dendrites overlapping by restricting the growth to closest neighbors only. This constraint was evaluated to be efficient for both task in Fig. 5 and task in Fig. 6.

Modifications

1. We extend the discussion on the adverse effect of multiple dendrites on the formation of a network: “Note that the time to bridge the bell and food electrodes is twice larger than the time to initially bridge the food and saliva electrodes. This effect corresponds to the growth of dendrites occurring only from the bell electrode dendrite. This mechanism could substantially increase the completion time during network formation but could be mitigating by engineering the electrode shapes and dendritic projection as in [8]”
2. We extend the discussion on hardware constraint for network mapping: see answer 1 to reviewer#1.

Q4: In Figure 5b, a shift from partial to full inter-neuron connections is observed. It would be interesting to know if the process could be reversed—from full connection to partial connection. Additionally, in Figure 5d, a decrease in accuracy after the number of connections reaches 30 in the structural plasticity SVM model raises the question of whether this phenomenon could be explained by overfitting. If so, how can we determine the suitable number of connections and when to stop training?

The possibility to reverse the electropolymerization remains a major challenge for this approach and electropolymerization in general. Once a dendrite is formed, it is indeed possible to modify its conductivity by two means: (i) using doping/de-doping strategies, as in electro-chemical memory transistors (see [7]). This effect can potentially be non-volatile but requires more complex circuit design and overhead. (ii) a simpler way to induce long term depression is to use over-oxidation of the PEDOT dendrites. We present in Fig. R2 the evolution of the current-voltage characteristic of a dendrite when the dendrite is over-oxidized by applying a large negative potential on an external gate (a Ag/AgCl global electrode in the electrolyte). This effect is non-volatile and non-reversible and could prevent the communication in between two nodes. Furthermore, as proposed in [3], over-oxidized polymer can present different stability and solubility in the electrolyte. An interesting option for our particular case would be to trigger dissolution of the over-oxidized form of PEDOT in water, while maintaining its stability in its standard configuration. This is the subject of current investigation by our group that we prefer to not include in the supplementary information, but will be the subject of future communications.

Figure R2: Overoxidation of PEDOT dendrites. A large negative potential on a global gate can over-oxidized the PEDOT and decrease its conductivity permanently.

Fig. 5b show a maximum of accuracy for a sparse network of 22 connections (13 structural connections and 9 analog connections for SVM). We present in Fig. S13 the evolution of the classification accuracy for both the train and test dataset. Overfitting can be evidenced when accuracy on training dataset increase or saturate while accuracy on testing dataset start to drop. Fig. S13 shows that the maximum of accuracy is observed for both training and testing dataset, thus rejecting the overfitting hypothesis. This highlight that an optimal sparse topology corresponding to an optimal projection of the signals exist for this task. Unfortunately, there is no good way of guessing the number of connections that would be optimal beforehand. Our train of thought is to let the network grow until a sufficient accuracy has been reached, if the accuracy hits a plateau or if it starts to decrease. Unlike L1 and L2 regularization, the network is sparse by default, and therefore more energy efficient.

Figure S13: Accuracy on the train and test set of the EMG dataset, ruling out the overfitting hypothesis.

Modifications

1. Addition of Fig. S13 and additional discussion in main text: “We estimate the network accuracy on both the training and testing dataset in Fig. S13.

The same maximum accuracy is observed on both dataset thus ruling out the hypothesis of overfitting by increasing the number of connections.”

Reviewer # 3

We thank the reviewer for their careful reading of the manuscript and comments. We propose below an additional discussion to address the reviewer’s suggestion.

The manuscript by Janzakova et al describes a novel implementation of the previously reported in-situ growth of PEDOT:PSS dendritic fibers. The authors extensively explain how structural plasticity is achieved and then proceed to highlight three demonstrative applications where this can be used. Although the concept is not new, the implementation is excellent, the applications are very relevant and impactful, and the experimental achievements are state-of-the art and very extensive. I believe this work can be published in Nature Communications. Perhaps the authors can add an outlook of how these systems can be scaled up in the future, or whether there are physical limitations. In traditional ANN there can be many layers and many hidden nodes and neurons. How would that be envisioned in these systems? A simple perspective / discussion would help.

We propose in the revised version an extended discussion about the hardware implementation potential of the proposed structural plasticity concept. This first point is addressed in the answer to reviewer#1, question1.

Nevertheless, the possibility to deploy structural plasticity on larger network, and deep-network in particular would be to find a way to create connections in between hidden layers. One of the key principles of Hebbian structural plasticity is that the presynaptic and postsynaptic node activity must be correlated for dendritic growth. In traditional ANNs, for connectivity to occur, there must be some kind of initial connectivity between the layers to have postsynaptic activity and enable dendritic growth. Otherwise, the input activity must be fed to all the layers to force some kind of initial activity. This is why in the EMG task the output neurons are stimulated with Poisson noise. In a multi-layer setting, it would be hard to decide which hidden neuron should fire and when as they are not directly associated with the output labels. It would be possible to have spontaneous activity to allow the creation of connections in a standard multi-layer ANN. This slightly reduces the Hebbian principle, and the effect on performance would need further study. An alternative option would be to consider layer-by-layer learning and to propagate the structural plasticity gradually. This mechanism could take inspiration from reward based STDP. For example. This option remains completely unexplored and is paving the way to future research directions in the exciting topic of network genesis.

Modifications

1. Additional discussion in conclusion: “One of the key principles of Hebbian structural plasticity is that the presynaptic and postsynaptic node activity must be correlated for dendritic growth. In traditional ANNs, for connectivity to occur, there must be some kind of initial connectivity between the layers to have postsynaptic activity and enable dendritic growth.

Otherwise, the input activity must be fed to all the layers to force some kind of initial activity. This is why in the EMG task the output neurons are stimulated with Poisson noise. In a multi-layer setting, it would be hard to decide which hidden neuron should fire and when as they are not directly associated with the output labels. It would be possible to have spontaneous activity in the hidden layers to allow the creation of connections in a standard multi-layer ANN. This would be some departure from the Hebbian principle, and the effect on performance would need further study.”

References

- [1] Matteo Cucchi et al. “Reservoir computing with biocompatible organic electrochemical networks for brain-inspired biosignal classification”. en. In: *Science advances* 7.34 (Aug. 2021). DOI: 10.1126/sciadv.abh0693.
- [2] Yuki Koizumi et al. “Electropolymerization on wireless electrodes towards conducting polymer microfibre networks”. en. In: *Nature communications* 7 (Jan. 2016), p. 10404. DOI: 10.1038/ncomms10404.
- [3] Jennifer Y Gerasimov et al. “An Evolvable Organic Electrochemical Transistor for Neuromorphic Applications”. en. In: *Advancement of science* 6.7 (Apr. 2019), p. 1801339. DOI: 10.1002/advs.201801339.
- [4] Kamila Janzakova et al. “Analog programming of conducting-polymer dendritic interconnections and control of their morphology”. en. In: *Nature communications* 12.1 (Nov. 2021), p. 6898. DOI: 10.1038/s41467-021-27274-9.
- [5] Matteo Cucchi et al. “Directed growth of dendritic polymer networks for organic electrochemical transistors and artificial synapses”. en. In: *Advanced electronic materials* 7.10 (Oct. 2021), p. 2100586. DOI: 10.1002/aelm.202100586.
- [6] Ankush Kumar et al. “Theoretical modeling of dendrite growth from conductive wire electro-polymerization”. en. In: *Scientific reports* 12.1 (Apr. 2022), p. 6395. DOI: 10.1038/s41598-022-10082-6.
- [7] Kamila Janzakova et al. “Dendritic Organic Electrochemical Transistors Grown by Electropolymerization for 3D Neuromorphic Engineering”. en. In: *Advancement of science* 8.24 (Dec. 2021), e2102973. DOI: 10.1002/advs.202102973.
- [8] Naruki Hagiwara et al. “Fabrication and training of 3D conductive polymer networks for neuromorphic wetware”. en. In: *Advanced functional materials* (June 2023). DOI: 10.1002/adfm.202300903.
- [9] Nikhil Garg et al. “Signals to Spikes for Neuromorphic Regulated Reservoir Computing and EMG Hand Gesture Recognition”. In: *International Conference on Neuromorphic Systems 2021*. ICONS 2021. Knoxville, TN, USA: Association for Computing Machinery, July 2021, pp. 1–8. DOI: 10.1145/3477145.3477267.

- [10] Y Ma et al. “Neuromorphic Implementation of a Recurrent Neural Network for EMG Classification”. In: *2020 2nd IEEE International Conference on Artificial Intelligence Circuits and Systems (AICAS)*. Aug. 2020, pp. 69–73. DOI: 10.1109/AICAS48895.2020.9073810.
- [11] Elisa Donati. *EMG from forearm datasets for hand gestures recognition*. May 2019. DOI: 10.5281/zenodo.3194792.

REVIEWERS' COMMENTS

Reviewer #1 (Remarks to the Author):

I thank the authors for their careful and insightful revisions. After reading the revised manuscript, I have to admit that my initial criticism was maybe mainly driven by my wish to see the scale-up of such systems. However, I agree with the authors that there is still much work to be done and the present study represents an important step towards more complex systems. So, I appreciate the work the authors spent on the discussion of the scale-up and taking the example of 2D networks. As a side note: The plasticity model used in this work will most likely be sufficient to describe the scale-up into more dense physical systems correctly as other plasticity rules (higher order) will come into play.

Anyway, I acknowledge the other modifications done by the authors (+ their additional comments) as they represent a straight and fair assessment.

minor point: please double-check the ms again. several references to figures are showing a ?? symbol.

Reviewer #2 (Remarks to the Author):

I thank the authors for the prompt responses, which have well addressed my previous questions. I have a minor suggestion that it could be valuable to discuss structural plasticity in the context of one-shot on-the-fly learning, specifically with regard to optical memristors, as this might be relevant. Overall, I recommend the revised paper for publication.

Reviewer #3 (Remarks to the Author):

The authors have successfully addressed all comments and the manuscript can now be accepted.

Reviewer # 2

I have a minor suggestion that it could be valuable to discuss structural plasticity in the context of one-shot on-the-fly learning, specifically with regard to optical memristors, as this might be relevant.

We want to thank you for this suggestion. We have added another discussion point in the manuscript, mentioning the beneficial aspect that optical memristors could bring when combined with structural plasticity in a one-shot/few-shot learning setting.

Modifications

1. Additional discussion: While structural plasticity requires some amount of data to connect neurons together, the lower parameter count of the resulting network should enhance its ability to learn with fewer data samples. One-shot or few-shot synaptic plasticity techniques could go well with this approach to enable devices to be trained at low energy and latency cost on-the-fly. Combined with other biocompatible devices such as optical memristors [1], such fast-learning systems could be implemented and trained directly in vivo.

References

- [1] Nathan Youngblood et al. “Integrated optical memristors”. en. In: *Nature photonics* 17.7 (May 2023), pp. 561–572. DOI: 10.1038/s41566-023-01217-w.